

# Anomaly prediction of Internet behavior based on generative adversarial networks

XiuQing Wang[1,2,3,*], Yang An[1,*] and Qianwei Hu[1]

[1] College of Computer and Cyber Security, Hebei Normal University, Shijiazhuang, Hebei, China
[2] Hebei Provincial Key Laboratory of Network & Information Security, College of Computer and Cyber Security, Shijiazhuang, Hebei, China
[3] Hebei Provincial Engineering Research Center for Supply Chain Big Data Analytics & Data Security, Hebei Normal University, Shijiazhuang, Hebei, China
[*] These authors contributed equally to this work.

## ABSTRACT

With the popularity of Internet applications, a large amount of Internet behavior log data is generated. Abnormal behaviors of corporate employees may lead to internet security issues and data leakage incidents. To ensure the safety of information systems, it is important to research on anomaly prediction of Internet behaviors. Due to the high cost of labeling big data manually, an unsupervised generative model–Anomaly Prediction of Internet behavior based on Generative Adversarial Networks (APIBGAN), which works only with a small amount of labeled data, is proposed to predict anomalies of Internet behaviors. After the input Internet behavior data is preprocessed by the proposed method, the data-generating generative adversarial network (DGGAN) in APIBGAN learns the distribution of real Internet behavior data by leveraging neural networks' powerful feature extraction from the data to generate Internet behavior data with random noise. The APIBGAN utilizes these labeled generated data as a benchmark to complete the distance-based anomaly prediction. Three categories of Internet behavior sampling data from corporate employees are employed to train APIBGAN: (1) Online behavior data of an individual in a department. (2) Online behavior data of multiple employees in the same department. (3) Online behavior data of multiple employees in different departments. The prediction scores of the three categories of Internet behavior data are 87.23%, 85.13%, and 83.47%, respectively, and are above the highest score of 81.35% which is obtained by the comparison method based on Isolation Forests in the CCF Big Data & Computing Intelligence Contest (CCF-BDCI). The experimental results validate that APIBGAN predicts the outlier of Internet behaviors effectively through the GAN, which is composed of a simple three-layer fully connected neural networks (FNNs). We can use APIBGAN not only for anomaly prediction of Internet behaviors but also for anomaly prediction in many other applications, which have big data infeasible to label manually. Above all, APIBGAN has broad application prospects for anomaly prediction, and our work also provides valuable input for anomaly prediction-based GAN.

Corresponding author
XiuQing Wang,
xqwang2013@163.com

## INTRODUCTION

With the development of enterprise information systems, a large number of business secrets, working data, and private data of employees and customers in the activities of production, operation, serving, and management are involved, and those secrets and data are valued as data assets by more and more enterprises. Although reliable systems are established in most organizations to prevent data leakage, cybersecurity incidents are still unavoidable. Abnormal behaviors of employees in enterprises are likely to cause sensitive data leakage (involving 60% of internet security issues and data leakage incidents are related to the misoperations and abnormal behaviors of employees (*CCF-BDCI, 2021*), and anomaly prediction of Internet behavior is critical for guarantee the security of information systems).

Anomaly detection focuses on finding non-conforming patterns of the expected behavior in the data. In various application fields, these non-conforming patterns are often referred to as anomalies, outliers, *etc.* (*Chandola, Banerjee & Kumar, 2009*). Anomaly detection is applied to various scenarios, such as detecting credit card fraud in the financial industry (*Ahmed, Mahmood & Islam, 2016*), identifying malicious buyers or sellers through transaction data in the e-commerce industry, detecting anomaly genes to identify lesions in biogenetics, detecting faults in security systems. Also, anomaly detection and prediction can also be applied for cybersecurity incident prediction and network intrusions identification (*Chandola, Banerjee & Kumar, 2009*).

With the increasing popularity of Internet-based information systems, users' behavioral data is becoming increasingly vast, and behavioral patterns are also becoming more and more intricate. Because of the large volume of Internet behavior log data, manual labeling of Internet behavior data is seriously costly and impractical in most cases. Consequently, anomaly prediction of behaviors based on data has become more critical. It is urgent for people to find effective methods to make anomaly predictions of Internet behaviors.

Unsupervised algorithms do not need labeled data, however, it is difficult to obtain satisfactory detecting results in large-scale, high-dimensional data sets by unsupervised methods when no outliers or outliers are scattered in the data set (*Smiti, 2020*; *Ning et al., 2022*). For example, the clustering result of the k-means algorithm is influenced by the K value, so the k-means algorithms can't reveal the intrinsic representation of data accurately due to noise. The Isolation Forest algorithm is only sensitive to global anomaly points, and it cannot isolate anomalies well for high-dimensional data. Deep learning (DL) is an important branch of machine learning that leverages the powerful nonlinear processing ability of deep neural networks (DNNs) to extract data features automatically. Moreover, DL's powerful nonlinear processing and the intrinsic data feature representing ability can also be used to realize transfer learning in various areas to achieve more extensive anomaly prediction. Generative adversarial networks (GAN) have been applied in computer vision, time series forecasting, and anomaly detection etc. successfully (*Goodfellow et al., 2014*). Various prediction and detection methods of GAN have demonstrated high performance in "Related Work", such as the anomaly prediction method–LogGAN proposed in *Xia et al. (2022)*, and so on.

A CCF Big Data & Computing Intelligence Contest (CCF-BDCI) is dedicated to solving anomaly prediction of Internet behavior, which is from real scenarios in government and enterprises (*CCF-BDCI, 2021*). We conduct research on Internet behavior data from CCF-BDCI, and propose a DL-based model–APIBGAN for anomaly prediction of Internet behavior using Generative Adversarial Networks (APIBGAN), which predict the outlier of Internet behaviors based on the employees' Internet behavior logs data, and the higher outlier indicates the higher potential risks. GANs have extensive applications across various domains, however, there is little existing literature on anomaly prediction of Internet behavior, except the methods based on Isolation Forest which from the CCF-BDCI literature (*CCF-BDCI, 2021*). To address the challenge of applying GAN to anomaly prediction of Internet behavior, our proposed APIBGAN using a small amount of online behavior data with outliers to train the generator of APIBGAN to generate online behavior data with outliers as a benchmark for anomaly prediction. As the input to the G is random noise, the Internet behavior data generated by the G also has some randomness in conforming to the original data distribution. This randomness indicates that GAN can learn the distribution of anomaly behavior data of employees instead of isolating the anomaly behaviors as some unsupervised algorithms, such as k-means and Isolation Forest. We also propose a data preprocessing method for feature fusion and selection of Internet behavior data. Combining the proposed data preprocessing method and the distance-based Internet behavior prediction method, APIBGAN achieves accurate anomaly prediction of Internet behavior by using a small amount of labeled behavior data and the Root Mean Square Error (RMSE) based metrics provided by CCF-BDCI. The final anomaly prediction scores, which are higher than that of the Isolation Forest (*CCF-BDCI, 2021*), demonstrate the effectiveness of APIBGAN in anomaly prediction of Internet behavior. The contribution of our work is as follows:

- We first propose an effective GAN-based APIBGAN model for anomaly prediction of Internet behavior without using a large amount of labeled data. Moreover, the structure of APIBGAN is simple, so it is easy to be implemented.
- We design a Data-generating GAN (DGGAN) for APIBGAN. By leveraging a small amount of labeled data, DGGAN learns the distribution of real data through gaming between the generator and the discriminator to generate samples with random noise as the input of APIBGAN. The generated samples can be used as the benchmark to realize anomaly prediction of Internet behavior by distance-based method.
- We propose a data preprocessing method–the label encoder-based method for Internet behavior data, which provides valuable input for the preprocessing of Internet behavior data.
- APIBGAN enhances data security and mitigates the risks of data leakage and data destruction stemming from improper behaviors. Except for anomaly prediction of Internet based information systems, APIBGAN can be applied in various fields, including the finance, e-commerce, and biogenetics industry.

The article is structured as follows: ''Related Work'' discusses the related works and introduces the principle of GAN. The methodology and implements of APIBGAN are

described in "GAN-based Anomaly Prediction of Internet Behavior". In "Experimental Results and Analysis", the details of experiments are described and the results are analyzed. "Discussion and Future Work" discusses the insights and the limitations of our work, and introduces the future work. Finally, "Conclusions" describes the conclusion of this work.

## RELATED WORK

### Review of outlier prediction

Anomaly prediction is a process of detecting outliers in a dataset, and anomaly detection algorithms are widely used in security, commercial and industrial applications. Many unsupervised algorithms and DL-based methods are widely used in anomaly detection. Next, we will review the algorithms based on unsupervised learning.

The distance-based methods are used for anomaly detection successfully. *Laurikkala et al. (2000)* investigated informal box plots to identify outliers in medical data of real-world, and the removal of outliers increased the classification accuracy of discriminant analysis functions and the nearest neighbor method. The work of *Zhang, Hutter & Jin (2009)* defined a local distance-based outlier factor (LDOF) to better discover outliers with high precision and good stability. *Jin et al. (2006)* proposed a simple but effective method for local outliers based on a symmetric neighborhood relationship. The proposed method considers neighbors and reverses neighbors of an object when estimating its density distribution so that outliers can be presented more meaningfully.

Many scholars devote to anomaly detection for high dimensional data, for example, *Rousseeuw & Van Driessen (2006)* proposed a method for fast-least trimmed squares (FAST-LTS) regression for large datasets to make the LTS estimator available as a tool for analyzing large data sets and detecting outliers. *Goldstein & Dengel (2012)* proposed an unsupervised Histogram-based Outlier Score (HBOS) algorithm for anomaly detection that records scores in linear time.

In addition, nonparametric machine learning methods are also employed for anomaly detection. *Latecki, Lazarevic & Pokrajac (2007)* proposed an unsupervised algorithm for outlier detection, which used a nonparametric kernel to estimate the density of data instances to detect outliers. *Kriegel, Schubert & Zimek (2008)* proposed a parameter-free Angle-based Outlier Detection (ABOD) method for high-dimensional data, which performs well on high-dimensional data.

With the rapid development of DL, DL-based methods can achieve better anomaly detection with the powerful nonlinear processing of DNNs. For example, in *An & Cho (2015)*, an anomaly detection method was proposed using reconstructed probabilities from a variational autoencoder, which is much more objective than using the reconstruction error of autoencoder and PCA-based methods. *Zong et al. (2018)* proposed a Deep Autoencoding Gaussian Mixture Model (DAGMM) for unsupervised anomaly detection, and their work suggests a promising direction for unsupervised anomaly detection on high-dimensional data. *Zhou et al. (2021)* proposed a deep support vector data description based on variational autoencoder (Deep SVDD-VAE) model to make anomaly detection with high precision.

## Review of GAN applied to anomaly detection

*Goodfellow et al. (2014)* proposed GAN in 2014, which is an unsupervised generative model, and is also one of the most promising DL methods. GANs could model complex high-dimensional distributions of real data from our life, which suggests that they can be effective for anomaly detection (*Zenati et al., 2018*). As a form of unsupervised learning algorithm, GAN has been widely used in anomaly detection (*Xia et al., 2022*).

In the anomaly detection of physical systems, *Li, Chen & Goh (2018)* constructed a GAN for LSTM networks that can handle temporal signals to detect anomaly behavior of physical systems. In *Hashimoto, Ide & Aritsugi (2021)*, GAN is used to detect visually undetectable anomalies successfully by analyzing multivariate time series of sensor data in semiconductor manufacturing. In the field of malware detection and malicious traffic detection, *Amin et al. (2022)* studied GAN' application for malware detection of the Android system. Intrusion Detection System GAN (IDSGAN), a GAN-based framework, was constructed to generate adversarial malicious traffic records to attack intrusion detection systems by spoofing and evading detection. The original malicious traffic records are converted into adversarial malicious records by the generator in *Lin, Shi & Xue (2018)*. A novel GAN-based IDS model detected unknown attacks using only normal data for in-vehicle networks in *Seo, Song & Kim (2018)*. *Rigaki & Garcia (2018)* use GAN to generate network traffic to simulate other types of traffic. In particular, the approach modified the network behavior of real malware to mimic the traffic of legitimate applications to avoid detection. *Fu et al. (2023)* and *Adiban, Siniscalchi & Salvi (2023)* used GAN to accomplish anomaly detection of network.

GAN-based anomaly detection methods are employed in other application domains, including out-of-domain (OOD) sample detection. *Marek, Naik & Auvray (2021)* proposed Out-of-Domain GAN (OODGAN) based on Sequence GAN (SeqGAN) to detect out-of-domain data generation. An effective adversarial attack mechanism-based GAN was proposed to augment hard OOD samples and a new generative distance-based classifier was designed to detect OOD samples instead of the traditional discriminator classifier with the threshold value in *Zeng et al. (2021)*. Furthermore, *Xia et al. (2021)* proposed a method named LogGAN using permutation event modeling GAN for anomaly detection, which detects log-level anomalies based on patterns. *Ibrahim et al. (2020)* proposed a new twist to generative models leveraging variational autoencoders as a source for uniform distributions to separate the inliers from the outliers.

The aforementioned methods achieve anomaly detection by learning the intrinsic representation of the data, and most of the referred methods are based on the classification of anomaly data. However, in the case of real-value anomaly prediction algorithms, it is necessary to learn the intrinsic representations of the data accurately to predict various outliers. Many applications require predictive models. Most approaches of anomaly detection or prediction use unsupervised algorithms, and the prediction of outliers is more challenging than anomaly classification. However, some researchers applied GAN in prediction tasks, such as the GAN-based method was used for tumor growth prediction in medical image processing by proposing a stacked 3D-GAN model named GP-GAN in *Elazab et al. (2020)*. Experimental results showed that GP-GAN achieves state-of-the-art in glioma growth prediction tasks. *Esmaeili et al. (2023)* and *Vyas & Rajendran (2023)* used

GAN to accomplish anomaly detection of medical image. *Chen et al. (2021)* introduced NM-GAN, an end-to-end pipeline that integrates an encode-decoder reconstruction network with a CNN-based discrimination network within a GAN-like architecture. *Lee & Seok (2021)* proposed a predictive probabilistic neural network model based on the conditional generative adversarial network (cGAN). By reversing the input and output of ordinary cGAN, the model can be used as a predictive model successfully. *Wang, Miller & Kesidis (2023)* proposed an unsupervised attack detector for DNN classifiers utilizing class-conditional GANs and modeling the distribution of clean data based on the predicted class label through an Auxiliary Classifier GAN (AC-GAN). *Contreras-Cruz et al. (2023)* developed the fast Anomaly Generative Adversarial Network (f-AnoGAN), a GAN architecture that exhibits superior accuracy when compared to bi-directional GAN and deep convolutional autoencoder. *Qin et al. (2023)* proposed a novel one-class classification-based ECG anomaly detection GAN. They innovatively integrated a bi-directional long-short term memory (Bi-LSTM) layer into the GAN architecture, and employed a mini-batch discrimination training strategy in the discriminator for synthesizing ECG signals. This approach facilitates the generation of samples that closely adhere to the data distribution of normal signals from healthy individuals, enabling the construction of a robust and generalized anomaly detector. *Meng et al. (2023)*, *Saravanan, Luo & Van Ngo (2023)*, and *Brophy et al. (2023)* also used GAN to accomplish the task of accurate time series prediction. *Ma et al. (2024)* used a multi-graph convolutional network (Multi-GCN) and gated recurrent unit (GRU) as the generator, revealing the hidden correlation between stock and stock time dependence. The aforementioned method demonstrates the advanced performance of GAN in prediction tasks and lays the foundation for using GAN for anomaly prediction of Internet behavior. However, there are few GAN-based methods for anomaly prediction of Internet behavior.

We propose APIBGAN to make anomaly prediction of Internet behavior. To learn a better data distribution, a data preprocessing method is designed for Internet behavior data and GAN is applied to anomaly prediction of Internet behavior and uses the data generated by the generator in GAN to predict the outliers. To facilitate the management of Internet behavior data, the outlier of Internet behavior data is defined by a number between 0 and 1 (*CCF-BDCI, 2021*). In this context, 0 represents normal behavior, while 1 represents abnormal behavior. The closer the outlier value is to 1, the higher the degree of abnormality of the Internet behavior, and the greater likelihood of privacy leakage. APIBGAN can also be applied in behavioral data from other domains for anomaly prediction, such as using credit card transaction information for antifraud of credit cards in financial field, using transaction data to detect "malicious buyers" in e-commerce industry, and detecting "lesions" through biological data in the biogenetic field, *etc.*, is important.

## Principle of GAN

A GAN consists of a generator $G$, and a discriminator $D$. $G$ and $D$ use the value function $V(D, G)$ to learn data representation. $G$ continuously fits the input noises $z$ to the real data to learn the distribution of real data. $D$ accurately discriminates whether the data is real or fake (real data is constructed the training set and fake data (generated data) is the data

generated by $G$). Through the continuous confrontation of the two models, $G$ eventually learns the probability distribution $p_g$ of the generated data, which is infinitely close to the probability distribution $p_{data}$ of real data. $V(D, G)$ is shown in Eq. (1),

$$\min_G \max_D V(D,G) = E_{x \sim p_{data}(x)}\big[\log D(x)\big] + E_{z \sim p_z(z)}\big[\log(1 - D(G(z)))\big], \tag{1}$$

where $E$ denotes the expected distribution of the data (*Goodfellow et al., 2014*), $D$ to maximize $V(D, G)$, which is to maximize the difference between the real data and the generated data, we call the generated data from $G$ fake data. The actual distribution $D(x)$ is to be close to 1, while the generated distribution $D(G(z))$ is to be close to 0. Furthermore, $G$ is to minimize the difference between $G(z)$ and the real data with $D(G(z))$ being close to 1. The schematic of GAN is shown in Fig. 1. Real and Fake are labels for the training $D$, Real is denoted as 1 for the real data and Fake is denoted as 0 for the fake data, respectively.

We solve for the optimal $D$ by maximizing $V(D, G)$. According to Eq. (1), we obtain Eq. (2). Maximizing $V(D, G)$ is also equal to maximizing the integration with Eq. (2) (*Goodfellow et al., 2014*),

$$\begin{aligned} V(D,G) &= E_{x \sim P_{data}}\big[\log D(x)\big] + E_{x \sim P_G}\big[\log(1 - D(x))\big] \\ &= \int_x P_{data}(x)\log D(x)dx + \int_x P_G(x)\log(1 - D(x))dx \\ &= \int_x \big[P_{data}(x)\log D(x) + P_G(x)\log(1 - D(x))\big]dx. \end{aligned} \tag{2}$$

When $D = D^\star$, there exists the maximum value of $V(D, G)$ ($D^\star$ is shown in Eq. (3)). The final training criterion $C(G)$ of $D$ is obtained by substituting $D = D^\star$, as shown in Eq. (4), using KL scatter to represent the distribution difference (*Goodfellow et al., 2014*).

$$D^* = \frac{p_{data}(x)}{p_{data}(x) + p_G(x)}, \tag{3}$$

$$C(G) = \max_D V(D,G) = -\log(4) + KL\left(p_{data}||\frac{p_{data} + p_g}{2}\right) + KL\left(p_g||\frac{p_{data} + p_g}{2}\right). \tag{4}$$

The criterion for training $D$ is to maximize the difference between the two distributions of $p_{data}$ and $p_g$, and that for training $G$ is to minimize the difference between the two distributions of $p_{data}$ and $p_g$. When $D^\star$ satisfies Eq. (3), the maximum value of $V(D, G)$ can be obtained, and the virtual training standard $C(G)$ reaches the minimum value $-\log 4$ if and only if $p_g = p_{data}$.

## GAN-BASED ANOMALY PREDICTION OF INTERNET BEHAVIOR

### Internet behavior data
#### Description and the division of the Internet behavior data
The effectiveness of APIBGAN is validated using the Internet behavior dataset, which is from the CCF-BDCI. This dataset contains 528,690 labeled Internet behavior data, and we divided the data into three groups: individual employees (Group 1), multiple employees

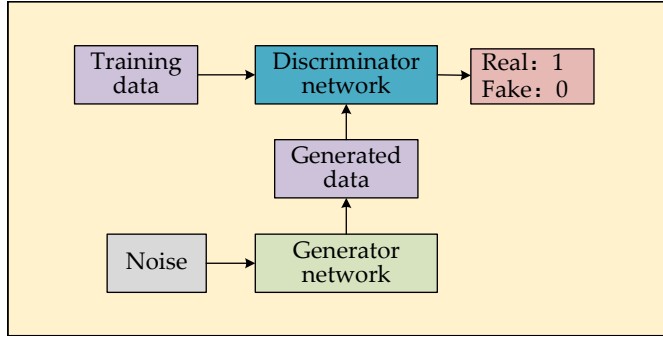

**Figure 1  Schematic of GAN.**

in the same department (Group 2), and multiple employees from different departments (Group 3). The number of employees in Groups 1 to 3 is becoming larger and larger, and the behavioral patterns of Groups 1 to 3 are becoming more and more complex. Such division is not only beneficial for training the APIBGAN to be suitable for various practical scenarios but also helpful in investigating the predictive performance of APIBGAN in various scenarios. The number of data in the three groups is 2,182, 19,472, and 528,690 respectively, the numbers of data in the training set for Group 1–3 are 600, 5,847, and 158,607 respectively, and those of the test set for Group 1–3 are 1,582, 13,625 and 370,083, respectively. Also, we divided the Internet behavior dataset into a training set and a test set at a ratio of 3:7 to train APIBGAN and evaluate the performance of APIBGAN. APIBGAN demonstrates its effectiveness with a smaller training set and a larger test set compared with the data division of most algorithms. The data of training set is labeled with outliers between 0 and 1 (*CCF-BDCI, 2021*), while the data of test set is unlabeled. Moreover, the details of the features of the Internet behavior data are showed in Table 1. The time feature of each data item records the time when employees access or log in to the Internet. Since many features of Internet behavior data are text type features, such as the URL and addresses are character, we only list three items of the Internet behavior data in Table 2 as page limitations and these features cannot be directly input into the DGGAN for training. After the data being preprocessed by our proposed algorithm, the processed data features are fed into GAN to learn the data distribution. The prediction methods based on distance realize the anomaly prediction of test data using generated samples.

### Analysis of the training set and the test set

The Mann-Whitney U test is conducted to investigate the distribution of the training set and the test set in Group 3 (the hypothesis testing results by SPSS are shown in Table 3), because the data in Group 3 has the largest volume of data and the most complex pattern. Moreover the detailed steps are as follows:

Denote $\sigma_A, \sigma_B$ as the distribution of the training set and the test set in Group 3, respectively. The null and the alternative hypothesis are shown in Eq. (5).

$$H_0 : \sigma_A = \sigma_B, H_1 : \sigma_A \neq \sigma_B. \tag{5}$$

**Table 1  Numbers and features of the Internet behavior data.**

| Feature | Number |
|---|---|
| id | 528,690 |
| account | 151 |
| group | 7 |
| vlan | 7 |
| IP | 133 |
| switchIP | 133 |
| port | 1,320 |
| url | 1,319 |

**Table 2  Details of the training set.**

| IP | URL | Port | Vlan | Switch IP | Time | ret |
|---|---|---|---|---|---|---|
| 192.168.1.50 | http://123.6.4.41 | 15788 | 700 | 129.30.06.37 | 2021/6/16 7:56 | 0.1149 |
| 192.168.31.46 | http://104.192.108.154 | 12665 | 700 | 152.91.85.45 | 2021/6/28 7:58 | 0.1801 |
| 192.168.2.3 | http://42.236.37.80 | 25551 | 700 | 129.30.06.37 | 2021/6/1 6:37 | 0.369 |

**Table 3  Significance of differences in the independent variables between the training and test sets.** At $\alpha$ ($\alpha = 0.05$) level of significance, the two-tailed Mann-Whitney U test is made to test the distribution of the training set and the test set in Group 3.

| Feature | p-value | Result |
|---|---|---|
| day | 0.315 | |
| hour | 0.257 | |
| minute | 0.733 | |
| account_switchIP | 0.055 | |
| account_IP_count | 0.851 | Reject $H_1$ |
| account_url_count | 0.521 | |
| account_switchIP_count | 0.902 | |
| account_url_IP_count | 0.994 | |
| url_IP_switchIP_count | 0.523 | |

According to Table 3, it can be concluded that the probability of significance of the distribution of each feature in both the training and test sets of Group 3 is greater than 0.05 at the significance level of 0.05, so we decline the alternate hypothesis $H_1$. According to the hypothesis testing results, we can draw such conclusion that the distribution of the training set in Group 3 are as the same as that of the test set.

## Problem description

Assume $X = [x_1, x_2, \cdots, x_n] \in R^{m*n}$ is the feature matrix of online behavior data without outliers, where $x_i \in R^n$ is the $i$-th online behavior data, $n$ is the number of online behavior data, and $m$ is the dimension of the feature for each online behavior data ($X$ is used as the test set). Assume $X^* = [x_1^*, x_2^*, \cdots, x_n^*] \in R^{(m+1)*n}$ is the feature matrix of online behavior data with outliers, where $x_i \in R^n$ is the $i$-th online behavior data, $n$ is the number of online

behavior data, $m +1$ is the feature dimension of the online behavior data with outliers, where the additional 1-dimensional feature is the outlier for each online behavior data ($X^*$is used as the training set). The outlier is between 0 and 1 (*CCF-BDCI, 2021*). The closer the outlier is to 1, the more anomalous the online behavior is; the closer the outlier is to 0, the more normal the online behavior is.

Manual labeling of online behavior data is not only costly, but also is impractical for a large volume of data. Though unsupervised algorithms can save human resources, their accuracy in anomaly prediction of Internet behavior is often low. To address this challenge, an unsupervised generative model, GAN based DL, is utilized to improve the accuracy of anomaly prediction. The proposed method uses a small number of online behavior data with outliers labeled manually to train the DGGAN to obtain the fake data. After processing the data with the proposed data preprocessing algorithm, we use the three-layer FNNs to construct the generator and the discriminator. Because of the gaming between the G and D in DGGAN, DGGAN is effective for generating Internet behavioral data (fake data) with outliers, although the structure of DGGAN is simple. Then, APIBGAN uses fake data to perform anomaly prediction in a large set of online behavior data without outliers. This paper contributes to generate Internet behavioral data with outliers using DGGAN by feature extraction of Internet behavior data, which are the benchmark for anomaly prediction of Internet behavior. APIBGAN uses the unsupervised generative model GAN to train data $X^*$ without outliers and to perform real value prediction of outliers on data $X$ with outliers to achieve anomaly prediction of Internet behavior. APIBGAN can also perform anomaly prediction for many application areas, such as anomaly prediction of a gene sequence in bioengineering and anomaly prediction of transaction records in the financial industry.

## Loss function

Since $D$ and $G$ in GAN are gamed, the gradient update of the parameters of $D$ and $G$ is performed through the loss function by Adam optimizer from *Kingma & Ba (2014)* in our work. Therefore, the loss functions Binary Cross-Entropy Loss (BCELoss) and Mode Seeking Loss (MSLoss), which are shown in Eqs. (6) and (8), are used to train DGGAN in APIBGAN.

BCELoss is a binary cross-entropy loss function for $D$ of the proposed method, and the loss is minimized when the actual label and output labels of $D$ are closest.

$$L_{\mathrm{BCE}} = -\frac{1}{N}\sum_{i=1}^{N}[y_i\log(p_i)+(1-y_i)\log(1-p_i)]. \qquad (6)$$

where $y_i$ is the actual label for data in $D$, we define the label for the real data as 1 and that for the fake data as 0, $p_i$ is the predicted label, and $N$ is the amount of training data. When the discriminator is trained, the real data is first input to $D$, $y_i$ is 1 at this time, so the loss is equivalent to the left part of Eq. (1). Then the fake data generated by $G$ is input to $D$, and $y_i$ is 0 at this time, so the loss is equivalent to the right part of Eq. (1).

MSLoss is the mode seeking loss in *Mao et al. (2019)*, which is applied to the generator in GAN to resolve the singularity of generated data.

The diverse data is generated by the generator when $L_{MS}$ is maximized. The distance between the generated data and that between the original noise is constrained by the distance function *Distance* so that the distance is maximized, which means that the local fit of the single pattern can be kept away so that the data output by the generator is more diversified. The absolute value-based Manhattan distance is used as the distance function for MSLoss, and the specific loss is shown in Eqs. (7) and (8),

$$\max_{G} V(Distance) = \max_{G} \left[ \frac{Distance(G(z_1), G(z_2))}{Distance(z_1, z_2)} \right]$$
$$= \max_{G} \left[ \frac{mean(abs(G(z_1) - G(z_2)))}{mean(abs(z_1 - z_2))} \right], \tag{7}$$

$$L_{MS} = \frac{1}{V(Distance) + \varepsilon}. \tag{8}$$

where $z_i$ is the $i$-th noise vector, $\varepsilon$ is a very small number to avoid a denominator of 0.

## Process of GAN-based anomaly prediction for Internet behavior
### Data preprocessing
We use label encoding in data preprocessing to convert text features of Internet behavior data to numeric ones. Then, the meaning of text features is turned into the definition of the numeric features correspondingly. It can lead to inconspicuous feature extraction if these features are directly input into the NN, because IP, URL and SwitchIP are different. So, the features of IP, URL and SwitchIP are spliced to get the advanced features. Table 4 shows the partial processed data and features of online behavior data corresponding to the data in Table 2, and the data in Table 4 is input into the DGGAN. The "_" in the table is the splicing of features. For example, "account_IP" is the feature of as a user using a specific IP. The "_count" in the table represents the count of the feature, for example, "account_IP_count" is the feature for the frequency of a user using a specific IP. Therefore, the different features are spliced together by "_" and "_count" to obtain advanced features representing the frequency of a user's online behavior. For example, the feature "url_IP_SwitchIP_count" represents the number of behaviors that use the IP to access the same URL through the SwitchIP. Through the training of DGGAN, the potential distribution of data can be learned accurately, and the data generated by $G$ contains behavioral patterns of the real data, so the generated samples can be used as a benchmark for predicting outliers accurately.

### Training of DGGAN
DGGAN is used to fit a small amount of training data with outliers so that the G can generate a large amount of Internet behavior data with outliers which represents the distribution of real Internet behavior data. The data generated by $G$ is used as a benchmark, and we can predict the outliers of the testing data by the benchmark generated by DGGAN. DGGAN uses FNNs as its structure, and the anomaly prediction based on Euclidean distance is employed to achieve anomaly prediction of Internet behavior. The detailed architecture and parameters of DGGAN are shown in Table 5. The more details and the source code for APIBGAN can be found at https://github.com/840962872/APIBGAN.

The training process of DGGAN is shown by Algorithm 1. Before the data are input to the GAN, normalization is needed to accelerate the fitting speed. The MinMaxScaler method in Sklearn is used to normalize the input data and then input it to the DGGAN for fitting training.

Algorithm 1: Training of DGGAN

1. **Input** $X = x_1, x_2, \cdots, x_n \in R^{(m+1)*n}$
2. **Init network** $G$ and $D$ with parameter $\theta_G, \theta_D$ using random parameter
3. Train network $f_{\theta_G}, f_{\theta_D}$ using the samples form $X$
4. **for** $i$ in $(1 : batchsize)$ **do**
    5. compute the label using $f_{\theta_D}(x_i)$
    6. compute the loss using BCELoss
    7. input noise $z$ into $f_{\theta_G}(z)$ to generate data
    8. obtain the label using $f_{\theta_D}(f_{\theta_G}(z))$
    9. calculate the loss using BCELoss and MSLoss
10. **end for**
11. update $\theta_G$ using $\theta_G = \theta_G + \alpha * \theta_G'$
12. update $\theta_D$ using $\theta_D = \theta_D + \alpha * \theta_D'$
13. **return** network model $G$ and $D$

## Anomaly prediction of internet behavior

APIBGAN aims to fit Internet behavior data with outliers by DGGAN. The trained G generates diverse Internet behavior data with outliers, and we use random noise as input of G to generate samples with uncertainty, which has a broader distribution than the used small real data. By setting the number of samples and inputting noise to the G, a fixed number of Internet behavior data can be generated, which defined as $data_G$. The test dataset of online behaviors without outliers is defined as $data_{X*}$. The outlier of $data_G$ is used as a benchmark to measure the similarity between $data_{X*}$ and $data_G$. The data of $data_G$ with the highest similarity to $data_{X*}$ is taken as the predicted outlier of $data_{X*}$. Distance-based methods can effectively discriminate the differences between benchmark and test set, so APIBGAN achieves anomaly prediction of Internet behavior. The prediction process of the outlier is shown in Algorithm 2, and the trained G is used as the model for predicting anomalies. Figure 2 shows the framework of APIBGAN, which contains two units, the DGGAN training unit and anomaly prediction unit. The DGGAN training unit mainly trains the DGGAN to generate Internet behavior data with outliers, which is also called generated sample. The anomaly prediction unit uses the Internet behavior data generated by the trained G (G is the trained model in the DGGAN training unit) as a benchmark to calculate the similarity of the testing Internet behavior data with the benchmark to achieve anomaly prediction.

**Table 4  Preprocessed data for training.**

| account_url__count | account_switchIP_count | account_url_IP_count | url_IP_switchIP_count | outlier[*] |
|---|---|---|---|---|
| 29 | 136 | 2 | 1 | 0.1149 |
| 10 | 308 | 1 | 1 | 0.1801 |
| 23 | 136 | 1 | 1 | 0.369 |

**Notes.**
*The outlier of Internet behavior data in CCF-BDCI.

**Table 5  Architecture and parameters of DGGAN.**

| Name | Networks Layer | Number of neuron | Optimizer | Learning rate |
|---|---|---|---|---|
| Discriminator | Input | 10 | Adam | 0.003 |
| | Latent1 | 256 | | |
| | Latent2 | 256 | | |
| | Output | 1 | | |
| Generator | Input | 4 | Adam | 0.003 |
| | Latent1 | 256 | | |
| | Latent2 | 256 | | |
| | Output | 10 | | |

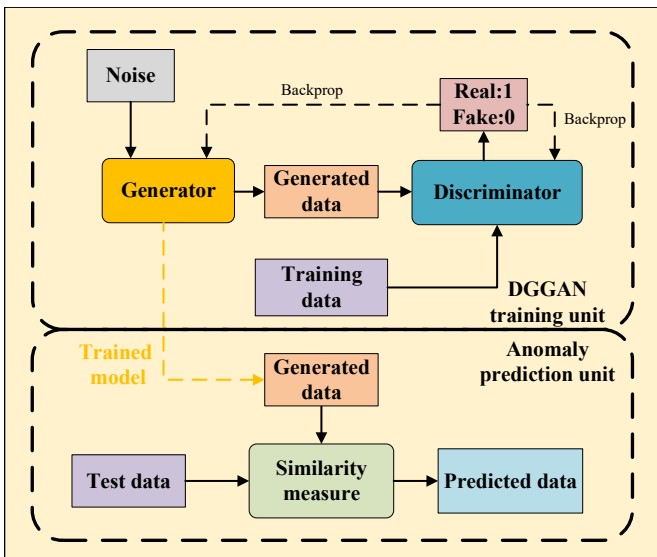

**Figure 2  Schematic of APIBGAN.**

Algorithm 2: Prediction of outliers

1. **Input** the noise $z = z_1, z_2, \cdots, z_n \in R^{4*n}$ into $G$ network
2. Generate $data_G$ for $n$ Internet behaviors with outliers
3. **Input** $data_{X*}$ without outliers and initialize matrix $M$ of Euclidean distance
4. for $m$ in $data_{X*}$ **do**
   5. for $n$ in $data_G$ **do**
      6. Calculate the Euclidean distance between $m$ and $n$
      7. Store distance into $M$
      8. The outlier value of $n$ from the minimum position in $M$ is the abnormal value of $m$
   9. **end for**
10. **end for**
11. **return** prediction $data_{X*}$

## Evaluation Metrics for anomaly prediction of Internet behavior data

RMSE is a commonly used metric to evaluate the deviation of predicted values from the actual values of a regression model, and the metrics based on RMSE are used in CCF-BDCI to evaluate the performance for anomaly prediction of Internet behavior. For this reason, we use RMSE-based metrics to evaluate the performance of APIBGAN for anomaly prediction of Internet behavior. The defined baseline result, which has the highest score in CCF-BDCI, utilizes an unsupervised method based on Isolation Forests for anomaly prediction over the entire data. However, the other unsupervised methods used in CCF-BDCI are not demonstrated except the method based on Isolation Forests in *CCF-BDCI (2021)*.

To evaluate the proposed method of anomaly prediction, we use a score to measure the accuracy of prediction. The *Score* provided by CCF-BDCI is calculated based on RMSE as Eqs. (9) and (10),

$$RMSE = \sqrt{\frac{\sum_{i=1}^{n}(X_{real} - X_{predict})^2}{n}}, \tag{9}$$

$$Score = \frac{1}{RMSE + 1}. \tag{10}$$

where $X_{predict}$ is the predicted outlier, $X_{real}$ is the true outlier, and $n$ is the number of test sets.

## EXPERIMENTAL RESULTS AND ANALYSIS

### Experimental results

We have conducted three experiments on Internet behavior data provided by CCF-BDCI, and each experiment used different Internet behavior data. The labeled data is used as training data and the unlabeled data is used as test data. We divide the Internet behavior data into three groups: Group 1 includes the online behavior data of an individual in a department, and the data of multiple employees in the same departments and different

**Table 6** Scores of anomaly prediction (100 samples and 1,000 samples).

| Data* | Score of 100 samples | Score of 1,000 samples | Score of the baseline** |
|---|---|---|---|
| Group 1 | 0.87230913 | 0.87277953 | |
| Group 2 | 0.83775343 | 0.85129635 | 0.81353900 |
| Group 3 | 0.82936291 | 0.83470446 | |

**Notes.**
*Three types of Internet behavior data.
**The highest Score in CCF-BDCI.

departments are Group 2 and Group 3. In the experimental results, the predictive accuracies are retained to eight decimal places.

The scores of APIBGAN using 100 and 1,000 generated data by G for Group 1 are 0.87230913 and 0.87277953, for Group 2 0.83775343 and 0.85129635, and for Group 3 0.82936291 and 0.83470446, respectively, which are shown in Table 6, respectively. Besides, the best score of Isolation Forest in CCF-BDCI is 0.81353900 (*CCF-BDCI, 2021*), and the scores of APIBGAN are higher than that of the method based on Isolation Forest in *CCF-BDCI (2021)*. From the experimental results we can see that the prediction accuracies of the data sets from the three groups have increased by 0.05%, 1.62%, and 0.6%, respectively with the generated data increased from 100 to 1,000. With the increasing amount of the generated data, we obtain higher prediction accuracies for the test sets, because GAN in APIBGAN uses random noise to generate various samples with the inherent uncertainty for both normal and abnormal Internet behavior, and a large volume of the generated samples broadens the distribution of the generated data. As the volume of generated data increases, the GAN generates diverse samples with the distribution of the real data. For 100 generated data, the accuracies of Group 1,2 are higher than that of Group 3 with 5.18% and 1.01%. For 1000 generated data, the accuracies of Group 1,2 are higher than that of Group 3 with 4.56% and 1.99%. With the Internet behavior data becoming complex from Group 1 to Group 3, it becomes hard for the G with the same structure to learn the high-level features of complex behavior data, and results in prediction accuracy decreasing. Even though, APIBGAN still makes anomaly prediction of Internet behavior effectively due to the inherent uncertainty of the generated samples.

From our experimental results and the best results in CCF-BDCI, we can draw such conclusion that it is effective to apply GAN to anomaly prediction of Internet behavior. In the future, we will research on new network structures to achieve anomaly prediction of Internet behavior, such as constructing cGAN-based networks for both the generator and discriminator to achieve anomaly prediction of Internet behavior.

The anomaly prediction scores of Internet behavior exhibit slight variations due to the different noises input into the generator in each experiment. We conducted 20 experiments to assess the performance of APIBGAN in predicting outliers compared to the baseline. The results of 20 experiments are presented in Fig. 3. As depicted in the figure, we can draw the conclusion that the APIBGAN method consistently outperforms the baseline across all 20 anomaly prediction scores of Internet behavior, despite some observed fluctuations.

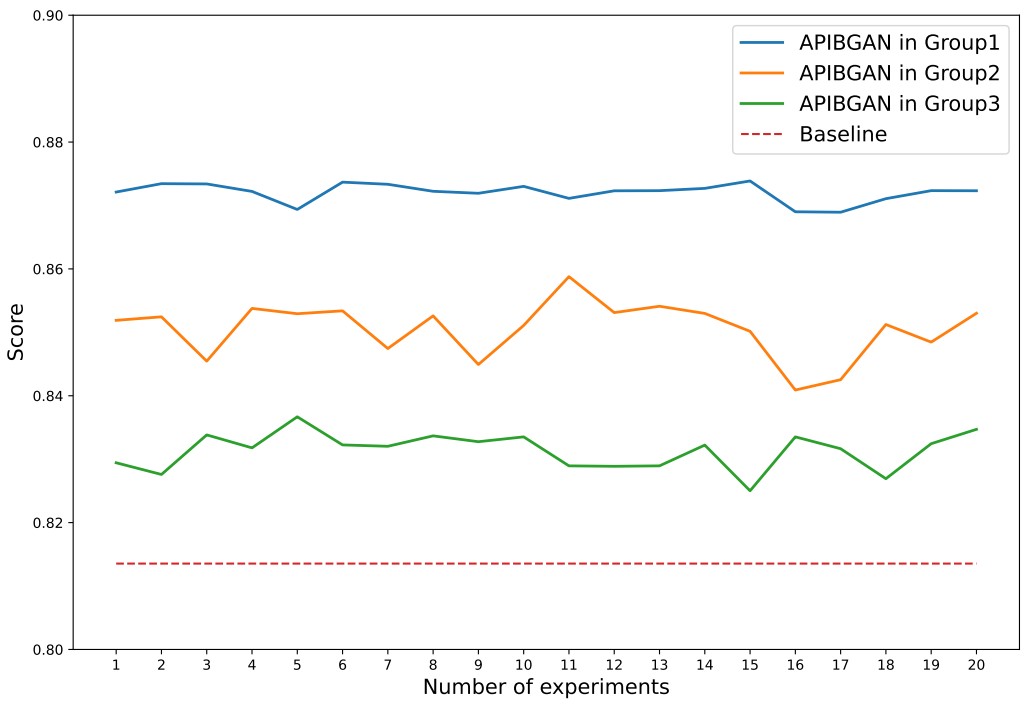

**Figure 3** Comparison of APIBGAN and baseline.

## Analysis for the experimental results

To explore how well the GAN fits the data, the probability density distributions of the actual and predicted outliers are visualized in Fig. 4. The same generative network model is used for generating data, the number of generated data in Figs. 4A and 4B is 100 and 1,000, respectively. The experimental results show that the prediction accuracy of small-scale and simple datasets is more accurate than large-scale and complex datasets, because the simple & small-scale data only represent specific behaviors, and the distribution of them is easy to be learned by DGGAN. The characteristics of complex & large-scale datasets are intricate, and it is difficult for DGGAN to accurately fit their Internet behavior features. However, the higher accuracy is achieved by increasing the number of the generated samples with higher computing cost. From the above analysis, we can draw the conclusions that it is effectively to apply the GAN in anomaly prediction of Internet behavior data. The proposed method can achieve higher accuracy than that of an unsupervised algorithm and predict the outliers of Internet behavior better.

## DISCUSSION AND FUTURE WORK

APIBGAN model uses a small amount of real Internet behavior data with outliers (30% of the total labeled Internet behavior data provided by CCF-BDCI) as training data, DGGAN learns the distribution of the real data only using three-layer-FNNs as G and D, and the trained G generates samples for anomaly prediction of Internet behavior. After that, we use a distance-based method to calculate the Euclidean distance between the samples

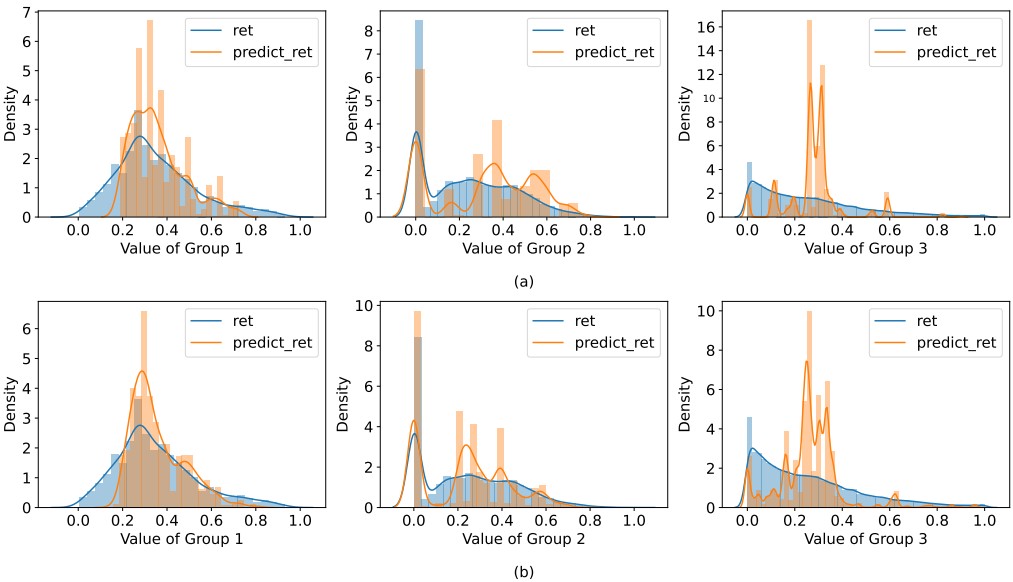

**Figure 4** **Probability density plot of predicted and true values.** (A) Probability density diagram of outliers based on 100 online behavior data generated by the generator. (B) Probability density diagram of outliers is based on 1,000 online behavior data generated by the generator. The results of three datasets from left to right, respectively, and the corresponding data were compared under the same model.

and the testing data to make anomaly prediction of Internet behavior. APIBGAN uses unsupervised generative models GAN to generate Internet behavior data with outliers as the benchmark to realize anomaly prediction of Internet behavior, and it is different from the classification-based algorithms in related works.

Through hypothesis testing for Internet behavior data, we can conclude that the distribution of the training set is equal to the distribution of the test set. Furthermore, the experimental results show that the prediction accuracy improves as the number of generated samples increases.

Compared with the experimental results of the unsupervised methods provided by CCF-BDCI, APIBGAN has better prediction accuracy, due to the powerful learning ability of DGGAN composed of DNNs. APIBGAN is effective for anomaly prediction of Internet behavior, although the structure of APIBGAN is simple, in which DGGAN consists of three-layer FNNs. Moreover, APIBGAN is easy to be implemented without high labor in manually labeling data.

Both APIBGAN and the method in *Lee & Seok (2021)* exploit the uncertainty of generated samples, while the purpose of APIBGAN is to apply GAN in anomaly prediction of Internet behavior, and APIBGAN combines the proposed data preprocessing method and distance-based anomaly prediction method to realize accurate anomaly prediction of Internet behavior. However, the proposed algorithm in *Lee & Seok (2021)* focuses on improving the structure of GAN by reversing the input and output, which makes the methods based on cGAN suitable for classification and prediction. Because of the high performance of the algorithm based on cGAN, we will combine the research fruits of the work in *Lee &*

*Seok (2021)* with our proposed method to design a novel and effective model for anomaly prediction of Internet behavior in the future.

We will also improve the structure of DGGAN to enable APIBGAN to learn intrinsic representations of more complex behaviors, such as utilizing convolutional neural networks (CNNs) to extract spatial features of behaviors. Improving the learning method of DGGAN, *e.g.*, cGAN can be used to generate more accurate samples with uncertainty by adding information related to the real data to the input of the generator.

The noise is crucial as it affects the generated Internet behavior data by G. We will consider using different types of noise to assess the performance of DGGAN in future work, such as incorporating the features of the original real data into the noise to ensure DGGAN learns a more accurate distribution of online behavior data. Besides, we will add certain constraints to the input noise for G to generate stable Internet behavior data with outliers. By imposing reasonable limitations to the random noise entry, the generated data is more likely to align the underlying patterns and structures of the real data, which can ultimately lead to more accurate and reliable anomaly detection for Internet behavior.

In addition, future work should therefore include putting forward a uniform anomaly prediction method based on data for various areas. It is worth exploring the application of APIBGAN in anomaly prediction for other areas, such as finance, medicine, and biology. In summary, investigating APIBGAN in various areas could provide valuable insights and potentially lead to new applications.

## CONCLUSIONS

Abnormal online behaviors of employees in enterprises are likely to cause sensitive data leakage and loss of data assets, so we propose an APIBGAN GAN-based model to make anomaly prediction of Internet behaviors with a small amount of labeled data, which structure is simple and easy to be implemented. In addition, experimental results and statistical analysis show that the data generated by G in APIBGAN represents the distribution of the real Internet behavior data well, and the APIBGAN is effective for anomaly prediction of Internet behaviors. Moreover, APIBGAN has better prediction accuracy than that of the comparison baseline method. To our knowledge, APIBGAN is the first study to apply GAN for anomaly prediction of Internet behavior, and our work also provides valuable input for GAN-based anomaly prediction. In the future, we can also use APIBGAN for anomaly prediction in other areas with a large volume of data infeasible to label manually, including the financial fields, e-commerce industry, biogenetic fields, security systems, Internet-based information systems, *etc.*, which have similar characteristics as that of anomaly prediction of Internet behavior.

### Funding

This work was supported by National Natural Science Foundation of China under contract (No.61175059, No. 61673160, and No. 61572170), Natural Science Foundation (NFS) of Hebei Province under contract (No. F2018205102), Foundation of Science and Technology Project of Hebei Education Department (No. ZD2021063), Key Foundation of Hebei Normal University (No. L2019Z11). The funders had no role in study design, data collection and analysis, decision to publish, or preparation of the manuscript.

### Grant Disclosures

The following grant information was disclosed by the authors:
National Natural Science Foundation of China: No. 61175059, No. 61673160, No. 61572170.
Natural Science Foundation (NFS) of Hebei Province: No. F2018205102.
Foundation of Science and Technology Project of Hebei Education Department: No. ZD2021063.
Key Foundation of Hebei Normal University: No. L2019Z11.

### Competing Interests

The authors declare there are no competing interests.

### Author Contributions

- XiuQing Wang conceived and designed the experiments, authored or reviewed drafts of the article, and approved the final draft.
- Yang An conceived and designed the experiments, performed the computation work, prepared figures and/or tables, authored or reviewed drafts of the article, and approved the final draft.
- Qianwei Hu performed the experiments, analyzed the data, prepared figures and/or tables, and approved the final draft.

### Data Availability

The Internet behavior data is available at GitHub and Zenodo:

- https://github.com/840962872/APIBGAN

- An Yang. (2024). 840962872/DatasetDataset: Internet behavior dataset (APIB-GAN_dataset). Zenodo. https://doi.org/10.5281/zenodo.10991245

The CCF-BDCI data, provided by Beijing Wondersoft Technology Co,. Ltd., is available at https://www.datafountain.cn/competitions/520/datasets.

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
