# Peer review of "Anomaly prediction of Internet behavior based on generative adversarial networks"

_PeerJ Computer Science, doi:10.7717/peerj-cs.2009_

## Round 0.1 · original submission · Major Revisions

Manuscript received review comments indicating the revisions on the presentation, technicality and experimental analysis. Comments are very constructive and I believe that they will help to strengthen the work. Hence authors are encouraged to submit the revised manuscript with revisions as per the suggestions.

Reviewer 1 has suggested that you cite specific references. You are welcome to add it/them if you believe they are relevant. However, you are not required to include these citations, and if you do not include them, this will not influence my decision.

Reviewer 1 ·

Basic reporting

Abstract
 The abstract does a reasonable job summarizing the research's key elements. However, the specific role of the GAN (Generative Adversarial Networks) and how it improves upon other methods could be better articulated. Clarify how your proposed method (APIBGAN) is distinctive and beneficial in the abstract.
 While the prediction accuracy is reported, it would be useful to include a comparison against the baseline or other existing methods in the abstract to highlight APIBGAN's effectiveness.

Introduction
 While the authors have cited references for anomaly detection applications, no references were mentioned to justify the selection of GANs for anomaly prediction or to provide background on other existing methods.
 The authors should further elaborate on the problems with current techniques and provide compelling reasons for using GANs, specifically APIBGAN, for anomaly prediction. How does APIBGAN specifically address the limitations of unsupervised methods mentioned in lines 90-96?
 In the introduction, authors could expand on the advantages of using APIBGAN, including its improvements in detection accuracy, its unique features, or how it addresses the shortcomings of other anomaly prediction techniques.
 While the authors have articulated the contributions of the study in lines 120-125, they could also emphasize the broader implications of their work, such as enhancing data privacy protection or improving detection of malicious activities.

Experimental design

Although the paper presents the mathematical model of APIBGAN, it does not explain it in plain language, which could limit its accessibility to readers not familiar with the field. Simplifying this language or adding an illustrative diagram could enhance understanding.
 While the authors mention that other unsupervised learning methods have low accuracy, there is no quantitative comparison. Including specific accuracy values of other methods compared with APIBGAN's results could strengthen the argument.
 The authors should clarify why outliers are selected between 0 and 1, and why the closer the outlier is to 1, the more anomalous the online behavior is. Is this based on a particular scale or a standard?
 As you have outlined your methodology, I notice a similarity with the methodology discussed in this paper: https://doi.org/10.3390/s21186194 It would be beneficial for readers if you could explicitly discuss the differences between your approach and the above paper, and justify how your method enhances or alters the findings.
 The usage of DGGAN for data generation is described but there is no comparison of DGGAN with other possible data generation models. Providing reasons for choosing DGGAN would strengthen the methodological rigor.
 While the increase in accuracy from 100 to 1000 generated data is mentioned, the paper could delve deeper into why this might be the case. What characteristics of the GAN make it more accurate with more data? Also, when you discuss the success of APIBGAN on smaller or single data sets, it would be beneficial to expand on why this might be the case.

Validity of the findings

The experiments are well-outlined but you could clarify more on why these particular datasets were chosen, what the rationale was behind the training-test split, and why these particular number of users were included in each set. Moreover, explaining the choice of using RMSE as the evaluation metric would enhance the comprehensibility of your methodology.

Additional comments

Overall Considerations
 The manuscript would benefit from a clear description of the datasets used, how they were prepared, and how the evaluations were conducted.
 The authors should propose possible improvements or extensions to their method, or potential applications that could be explored in the future.
 There is inconsistency in the use of abbreviations for "Figure". In some parts, you use "Fig." and in others "Figure". For clarity and consistency, it would be helpful to standardize this throughout the manuscript.
 To ensure reproducibility of the research, it would be helpful if the authors could provide further details about the architecture of APIBGAN, hyperparameter settings, and training methods.

Cite this review as

Reviewer 2 ·

Basic reporting

The authors proposed an unsupervised generative model-APIBGAN to predict anomalies in online behaviors, and the work of the paper is interesting and significant in practical use. The structure of the proposed abnormal behaviors detection model is simple and easy to be implemented.

Experimental design

Extensive experiments have been designed and performed.

Validity of the findings

The experimental results have validated the effectiveness of the proposed method ( the source code of the proposed method also be presented on https://github.com/840962872/APIBGAN.) .

Additional comments

I have some questions to strengthen the manuscript to meet the needs of high quality Journal - Peerj Computer scinece.
1. In order to let people to use the proposed model in practice, more details of the proposed model should be presented, such as how the APIBGAN model is trained and the detailed parameters of the trained APIBGAN model and so on.
2. Figure 3 is not clear enough and the resolution of Figure 3 needs to be increased.
3. The author should pay to the consistency of the paper, for example, in the hypothesis testing section, the format of H0 in the body text and in the formula should be consistent (H0 or H0).
4. The symbol "D" in Algorithm2 conflicts with the abbreviation D of Discriminator and should be adjusted.
5. The authors should pay attention to the format of the paper:
1) Formulas are also part of the body text, and there is no need for a separate sentence when describing the symbols in the formulas. Equations need to be followed by adding "," or "." according to the corresponding sentence. For example, "," should be added after Eq. (1).
2) Although the overall organization is good, some text of the paper also needs polished and proof reading.

Cite this review as

Reviewer 3 ·

Basic reporting

The repetition of certain sentences, the lack of definition for the second contribution, grammatical mistakes, the need for better highlighting of the proposed model comparison, and the suggestion to use the dataset heading in the above data processing heading. These are valid points that can improve the clarity and quality of the paper. I will make sure to address these issues and make the necessary revisions to enhance the overall presentation and readability of the document.

Experimental design

The article briefly mentions the potential application prospects of Generative Adversarial Networks (GANs) for anomaly prediction in other data domains, such as financial data or biomedical data. However, it does not delve into the specific challenges or benefits of applying GANs in these domains. Expanding on this point would provide readers with a better understanding of the broader implications of the research.

While the article mentions the prediction accuracy of the three studied cases, it does not provide any information on the evaluation methodology or comparative baselines. To improve the article, it would be beneficial to include a description of the evaluation metrics used, the size of the labeled dataset, and any other relevant details to help readers understand the significance of the achieved accuracies.

Validity of the findings

The article briefly mentions that the three-layer Fully Connected Neural Network (FNN) is effective in predicting internet behavior outliers, but it does not explain why or provide any analysis or interpretation of the results. To enhance the article, it would be valuable to include a discussion on the factors contributing to the model's effectiveness and potential limitations or areas for improvement.

Additional comments

The article should provide a stronger justification for studying the internet behavior data of employees in a company. This could include discussing the potential risks associated with improper internet behavior and the need for effective anomaly detection techniques to mitigate those risks.

The article should provide a stronger justification for studying the internet behavior data of employees in a company. This could include discussing the potential risks associated with improper internet behavior and the need for effective anomaly detection techniques to mitigate those risks.


The article lacks a clear introduction that establishes the significance of anomaly detection in internet behavior data. It would be beneficial to provide a brief overview of the importance of identifying non-conforming patterns in various application fields. Additionally, the article should provide a more comprehensive background on the challenges of protecting sensitive data in the context of internet behavior analysis.

Cite this review as

---

## Round 0.2 · Major Revisions

The manuscript is reviewed now. You would see that reviewers are suggesting the revisions in terms of the comparison with literature, justification, etc. Please revise the manuscript as per the suggestions and submit the revised manuscript along with the list of corrections carried out.

Reviewer 1 ·

Basic reporting

The authors modified the manuscript by appropriately reflecting the previous comments.

Experimental design

The authors modified the manuscript by appropriately reflecting the previous comments.

Validity of the findings

The authors modified the manuscript by appropriately reflecting the previous comments.

Additional comments

The revised manuscript presents an innovative approach towards addressing the significant challenge of anomaly prediction in internet behavior, especially in the context of corporate security. The approach, APIBGAN, leverages a Generative Adversarial Network framework, demonstrating proficiency in handling internet behavior data with minimal labeled data. This is particularly commendable given the inherent complexities and the high-dimensional nature of such data.

The authors have done a commendable job in structuring the manuscript. The introduction effectively sets the stage for the research, highlighting the relevance and the urgency of the problem in the backdrop of increasing cybersecurity threats. The detailed description of the methodology, supported by relevant literature, provides clarity on the proposed model's design and functioning. The incorporation of a Data-generating GAN (DGGAN) within the APIBGAN model is a novel approach that contributes significantly to the field of unsupervised learning in anomaly detection.

In summary, the revised manuscript is well-written, methodologically sound, and contributes significantly to the field of anomaly prediction using deep learning. The research holds substantial potential for practical applications in safeguarding internet-based information systems against anomalous behaviors, thereby enhancing data security in various industries.

Cite this review as

Reviewer 3 ·

Basic reporting

I have the following concerns; please address it before finalizing the draft.

The methodology described above has a limitation because it does not compare the proposed APIBGAN model with other existing methods for anomaly prediction of Internet behavior. The paper mentions that there is little literature on this topic, except for methods based on isolation forests. However, it does not thoroughly evaluate and compare the APIBGAN model with other methods, including Isolation Forest, to determine its effectiveness and performance in predicting anomalies.
The paper does not mention the size of the dataset used to train and test the APIBGAN model. It is important to consider the dataset size and its representation of real-world scenarios to ensure that the proposed model can be applied to different situations.

The paper does not discuss the potential limitations or challenges associated with using GANs for anomaly prediction of Internet behavior. GANs have their own limitations, such as mode collapse and training instability, which could affect the performance and reliability of the APIBGAN model.
The research proposes the use of the unsupervised generative model, APIBGAN, to avoid the high cost of manually labeling big data. However, this approach relies on a small amount of labeled data, which may limit the model's ability to accurately predict anomalies.

Experimental design

The research mentions that the input Internet behavior data is preprocessed using the proposed method, but it does not provide detailed information about the preprocessing steps. Without understanding the preprocessing method, it is difficult to assess the reliability and validity of the generated data.
The GAN in APIBGAN is composed of a simple three-layer fully connected neural network (FNNs). This simplistic architecture may limit the model's ability to capture complex patterns and nuances in the Internet behavior data, potentially affecting the accuracy of anomaly prediction.

A comparative analysis is often necessary for comparing existing problems and solutions, researchers can identify gaps and propose novel approaches or improvements. This analysis helps establish the context and significance of the research, highlighting its contribution to the field.

Validity of the findings

The research only provides prediction scores for three categories of Internet behavior data. While these scores are compared to a comparison method based on Isolation Forests, there is no information provided about other evaluation metrics, such as precision, recall, or F1 score. This limited evaluation makes it difficult to assess the overall performance of APIBGAN.
The research does not mention any external validation or replication of the results. Without external validation from other researchers or independent datasets, it is challenging to determine the generalizability and reliability of the findings.
The research claims that APIBGAN can be used for anomaly prediction in many other applications with big data. However, there is no empirical evidence or discussion provided to support this claim. It is unclear how well the proposed method would perform in different domains or datasets.
A sufficient number of references to support their claims and demonstrate a thorough understanding of the current literature. Including 70% of references from the last four years can be a good practice to ensure the paper reflects the most recent advancements in the field.

Cite this review as

---

## Round 0.3 · accepted · Accept

Reviewers have agreed with the revisions made in the manuscript as per their suggestions. We appreciate your efforts in revising the manuscript.

Reviewer 2 ·

Basic reporting

The description of the proposed method is clear, and the writing has been polished.

Experimental design

The experimental design of this paper is relatively reasonable, with rich experimental content, which can fully demonstrate the effectiveness of the proposed method in this paper.

Validity of the findings

The innovative description of this paper is relatively clear, and the conclusion section can provide a good summary of the paper.

Additional comments

The author has made improvements to the paper based on the comments of the reviewers, and has fully responded to their comments.

Cite this review as